# Inhibition of base editors with anti-deaminases derived from viruses

Zhiquan Liu [1,5], Siyu Chen[1,5], Liangxue Lai[1,2,3,4✉] & Zhanjun Li [1✉]

Cytosine base editors (CBEs), combining cytidine deaminases with the Cas9 nickase (nCas9), enable targeted C-to-T conversions in genomic DNA and are powerful genome-editing tools used in biotechnology and medicine. However, the overexpression of cytidine deaminases in vivo leads to unexpected potential safety risks, such as Cas9-independent off-target effects. This risk makes the development of deaminase off switches for modulating CBE activity an urgent need. Here, we report the repurpose of four virus-derived anti-deaminases (Ades) that efficiently inhibit APOBEC3 deaminase-CBEs. We demonstrate that they antagonize CBEs by inhibiting the APOBEC3 catalytic domain, relocating the deaminases to the extranuclear region or degrading the whole CBE complex. By rationally engineering the deaminase domain, other frequently used base editors, such as CGBE, A&CBE, A&CGBE, rA1-CBE and ABE8e, can be moderately inhibited by Ades, expanding the scope of their applications. As a proof of concept, the Ades in this study dramatically decrease both Cas9-dependent and Cas9-independent off-target effects of CBEs better than traditional anti-CRISPRs (Acrs). Finally, we report the creation of a cell type-specific CBE-ON switch based on a microRNA-responsive Ade vector, showing its practicality. In summary, these natural deaminase-specific Ades are tools that can be used to regulate the genome-engineering functions of BEs.

[1] Key Laboratory of Zoonosis Research, Ministry of Education, College of Animal Science, Jilin University, Changchun 130062, China. [2] CAS Key Laboratory of Regenerative Biology, Guangdong Provincial Key Laboratory of Stem Cell and Regenerative Medicine, South China Institute for Stem Cell Biology and Regenerative Medicine, Guangzhou Institutes of Biomedicine and Health, Chinese Academy of Sciences, Guangzhou 510530, China. [3] Guangzhou Regenerative Medicine and Health Guang Dong Laboratory (GRMH-GDL), Guangzhou 510005, China. [4] Institute for Stem Cell and Regeneration, Chinese Academy of Sciences, Beijing 100101, China. [5] These authors contributed equally: Zhiquan Liu and Siyu Chen. ✉email: lai_liangxue@gibh.ac.cn; lizj_1998@jlu.edu.cn

CRISPR-guided DNA base editors, which precisely install targeted point mutations without requiring DNA double strand breaks (DSBs) or donor templates, have exhibited a powerful genome manipulation capability in various organisms[1,2]. Cytosine base editors, consisting of a cytidine deaminase fused to a catalytically impaired Cas9 protein and one or more copies of uracil glycosylase inhibitor (UGI), generate C-to-T nucleotide substitutions in genomic target sites[3]. Similar to other Cas9-directed genome-editing tools, CBEs induce Cas9-dependent off-target (OT) mutations at off-target genomic loci that have high sequence homology to the target protospacer[3–5]. In addition, Cas9-independent deamination results in genome-wide Cas9-independent OT mutations and transcriptome-wide OT RNA mutations[6–9]. In addition, CBEs have been widely used in therapeutic applications, in which their activities often needs to be regulated in vivo[10,11]. To date, many phage-derived Acr proteins and small molecule-based CRISPR-Cas9 inhibitors have been reported[12–14], while there are no inhibitors that can specifically regulate deaminases, the main functional domains of CBEs. Developing deaminase off switches is necessary to modulate CBE activity.

Bacterial CRISPR-Cas systems utilize sequence-specific RNA-guided nucleases to defend against bacteriophage infection[12]. In response to the bacterial war on phage infection, numerous phages produce Acr proteins to block the function of CRISPR-Cas systems[12,13]. Similarly, viruses have evolved many natural anti-deaminase proteins through battles between viruses and APOBEC3 deaminases (Fig. 1a). The APOBEC3 family of proteins in mammals consists of cellular cytosine deaminases and well-known restriction factors against retroviruses[15,16]. As a countermeasure, viruses have evolutionarily acquired a series of genes to inhibit the antiviral activity of APOBEC3 proteins[16].

In this study, we repurpose deaminase-inhibiting proteins derived from viruses to inhibit base editors. We expanded the application of Ades to other types of BEs by rationally engineering deaminases. These Ades do not only inhibit Cas9-dependent OT activity but also dramatically decrease Cas9-independent OT activity. In addition, Ade1 was used to generate a cell type-specific CBE-ON switch based on a microRNA-responsive Ade vector. These Ades, together with existing inhibitors, strengthen the inhibitor toolbox for efficient regulation of BE activity in gene modification and therapeutic applications.

## Results

**Virus-derived anti-deaminases efficiently inhibit APOBEC3-CBEs.** The arms race between viruses and APOBEC3 deaminases has generated many natural anti-deaminases. We first selected seven Ades, including EBV-BORF2[17,18], KSHV-ORF61[17,19], HIV-1-Vif[20], SIVmac239-Vif[21,22], HSV-1-ICP6[19], EV71-2C[23] and HBV-HBx[24] (referred to as Ade1-Ade7), which originate from different viruses and have been reported to antagonize APOBEC3 deaminases in vitro (Table 1). Three APOBEC3-CBEs (A3A, A3B and A3G) were constructed with deaminase-nCas9-2xUGI architecture (BE4max, a state-of-the-art CBE)[25] (Fig. 1b). To test Ade inhibition of APOBEC3-CBEs, we transfected CBE and single guide RNAs (sgRNAs) (targeting EMX1-1 and FANCF) into HEK293T cells in the presence or absence of Ade proteins. As a result, four of seven Ades (Ade1-Ade4) showed obvious inhibition of C-to-T activities, while others showed no effect at either target site (Fig. 1c, d). Ade2 was the only effective inhibitor of A3A-CBE, showing slight inhibition, from 1.5- to 2.1-fold (Fig. 1c, d). Notably, Ade1 and Ade2 efficiently inhibited A3B-CBE, from 5.8- to 6.6-fold and from 2.7- to 2.9-fold, respectively (Fig. 1c, d). Both Ade3 and Ade4 exhibited strong inhibition of A3G-CBE, from 3.6- to 4.5-fold and from 3.6- to

4.9-fold, respectively (Fig. 1c, d). Moreover, we observed obvious dose-dependent inhibitory effects of Ade1 and Ade2 (Fig. 1e, f). However, Ade3 and Ade4 exhibited efficient inhibition at low doses, and an increase in doses did not further enhance the inhibition rate (Fig. 1g), which suggests that these Ades may function through two different inhibitory mechanisms. Additionally, we evaluated their inhibitory effect on three A3 deaminases from other species, including RmA3Bctd (rhesus monkey), mA3CDA1[26] (mouse) and SsA3Bctd[27] (Sus scrofa). These Ades did not show an inhibitory effect on the tested deaminases, except for Ade1, which slightly inhibited RmA3Bctd-CBE, suggesting that these Ades are species-specific and mainly evolved to inhibit human A3 deaminases[15] (Supplementary Fig. 1). These results suggested that the Ade1-Ade4 proteins can efficiently inhibit CBEs with differing APOBEC3 selectivity, holding the potential to be off-switches for CBE applications.

**Inhibitory mechanisms of Ades.** The A3A protein contains a single catalytically active cytidine deaminase domain. However, some A3 family members, such as A3B and A3G, have a C-terminal catalytic domain (CTD) and an N-terminal pseudo-catalytic domain (NTD) that retain the same tertiary folds but are not catalytically effective[15,28]. To evaluate whether Ades directly inhibit DNA deamination activity, we first constructed both A3Bctd-CBE and A3Gctd-CBE, which only contain the functional CTD (Fig. 2a). As a result, Ade1 and Ade2 exhibited the same inhibitory effect on A3B-CBE and A3Bctd-CBE, indicating that they may directly interact with A3Bctd and inhibit DNA deamination activity (Fig. 2b). In contrast, Ade3 and Ade4 completely abolished the inhibition of A3Gctd-CBE but not A3G-CBE, suggesting that they did not directly inhibit the deamination activity (Fig. 2b). Subsequently, western blot results indicated a remarkable decrease in A3G-CBE protein levels in the presence of Ade3 and Ade4, indicating that Ade3 and Ade4 inhibit A3G-CBE by activating its degradation (Fig. 2c, d). No obvious changes were observed in other A3-CBEs when the corresponding Ade was added (Supplementary Fig. 2). These results were consistent with previous reports showing that Ade1 interacted preferentially with A3Bctd in vitro and that Ade3 hijacked cellular proteasomal degradation pathways to degrade the whole A3G deaminase[17,20,29].

Ade1 inhibited only A3B-CBE, while Ade2 inhibited both A3A-CBE and A3B-CBE. To understand the different inhibition mechanisms of Ade1 and Ade2, we aligned the amino acid sequences of A3A, A3Bctd and A3Gctd (Supplementary Fig. 3a). The protein sequence of A3A is highly homologous with that of A3Bctd, but their key DNA binding loop 1 (L1) is quite different (Supplementary Fig. 3a). In addition, it has been reported that Ade1 interacts specifically with loop7 (L7) residues of A3Bctd in vitro[17]. The structural data also demonstrated that L1 together with L7 control substrate access to the open catalytic pocket of deaminase[30]. Therefore, we speculated that L1 and L7 might be the key regions controlling Ade inhibition of A3-CBEs. To validate our hypothesis, we constructed a series of A3-CBE chimeras by exchanging L1 and L7 in A3A, A3Bctd and A3Gctd (Supplementary Fig. 3b). As a result, Ade1 showed higher fold inhibition to A3A-L1B than to A3A (3.26-fold vs. 1.06-fold) and remarkable inhibition to A3Gctd-L1B compared with that to A3Gctd (5.67-fold vs. 0.96-fold). Its fold inhibition to A3Bctd-L1A was significantly decreased compared with that to A3Bctd (1.47-fold vs. 5.20-fold), suggesting that the L1 region of A3Bctd is indispensable for the inhibitory action of Ade1 (Fig. 2e and Supplementary Fig. 3c). In addition, Ade1 showed no inhibition of A3Bctd-L7G but significantly suppressed A3Bctd (1.06-fold vs. 5.20-fold). It also slightly improved the fold inhibition of A3Gctd-

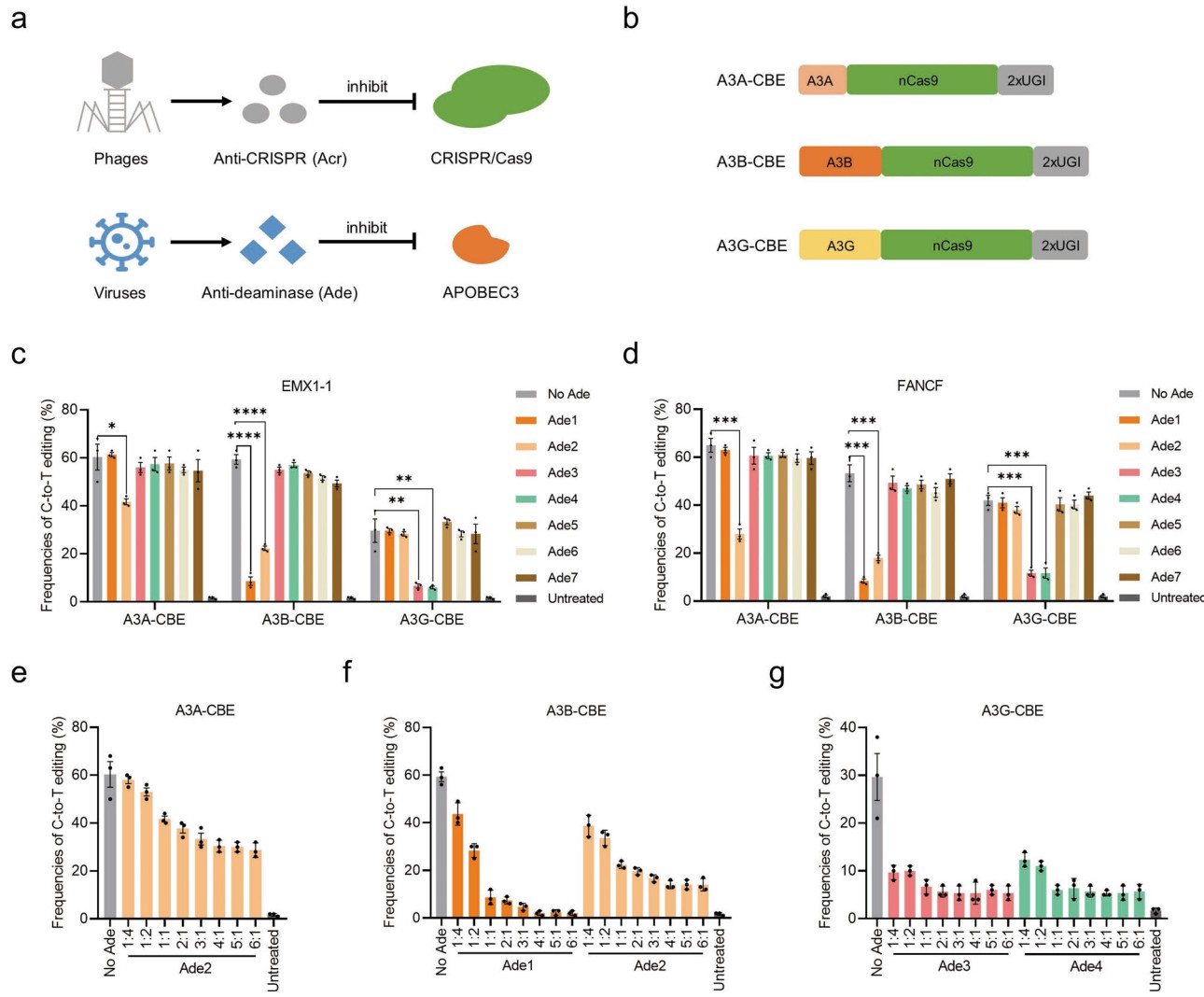

**Fig. 1 Viruses derived Ades efficiently inhibit A3-CBEs. a** Schematic representation of anti-CRISPR and anti-deaminase. **b** Schematic representation of three A3-CBE architectures. nCas9, D10A. **c, d** Base editing of A3A-, A3B- and A3G-CBE in the presence or absence of the seven Ades at the EMX1-1 (**c**) and FNACF (**d**) sites. Plasmids expressing CBE, sgRNA, and each Ade (1:1:1) were cotransfected into HEK293T cells. **e–g** Inhibition of A3A-CBE (**e**), A3B-CBE (**f**) and A3G-CBE (**g**) with different doses of Ades at the EMX1-1 site. The ratio of Ade:CBE ranging from 1:4 to 6:1. Values and error bars reflect the mean ± s.e.m. and $n = 3$ biologically independent experiments. All $p$ values were calculated by two-sided t tests. $*p < 0.05$, $**p < 0.01$, $***p < 0.001$, $****p < 0.0001$. Source data are provided as a Source Data file.

---

**Table 1 Summary of characteristics of the anti-deaminases (Ades) used in this study.**

| Ades | Viruses | Viral inhibitors | APOBEC3 proteins[1] | A3A-CBE[2] | A3B-CBE[2] | A3G-CBE[2] | Mechanism of inhibition[3] |
|------|---------|------------------|---------------------|------------|------------|------------|----------------------------|
| Ade1 | EBV | BORF2 | A3B | − | +++ | − | Inhibition of deaminase activity and relocalization |
| Ade2 | KSHV | ORF61 | A3A, A3B | + | ++ | − | Inhibition of deaminase activity and relocalization |
| Ade3 | HIV-1 | Vif | A3G | − | − | +++ | Degradation of deaminase |
| Ade4 | SIVmac239 | Vif | A3B, A3G | − | − | +++ | Degradation of deaminase |
| Ade5 | HSV-1 | ICP6 | A3A, A3B | − | − | − | NA |
| Ade6 | EV71 | 2 C | A3G | − | − | − | NA |
| Ade7 | HBV | HBx | A3G | − | − | − | NA |

[1]The APOBEC3 proteins inhibited by viral inhibitors were reported in previous studies.
[2]The APOBEC3-CBEs inhibited by Ades were determined in this study. −: no inhibition; +: slight inhibition; ++: moderate inhibition; +++: strong inhibition.
[3]The mechanism of inhibition was demonstrated in this study.
*NA* not applicable.

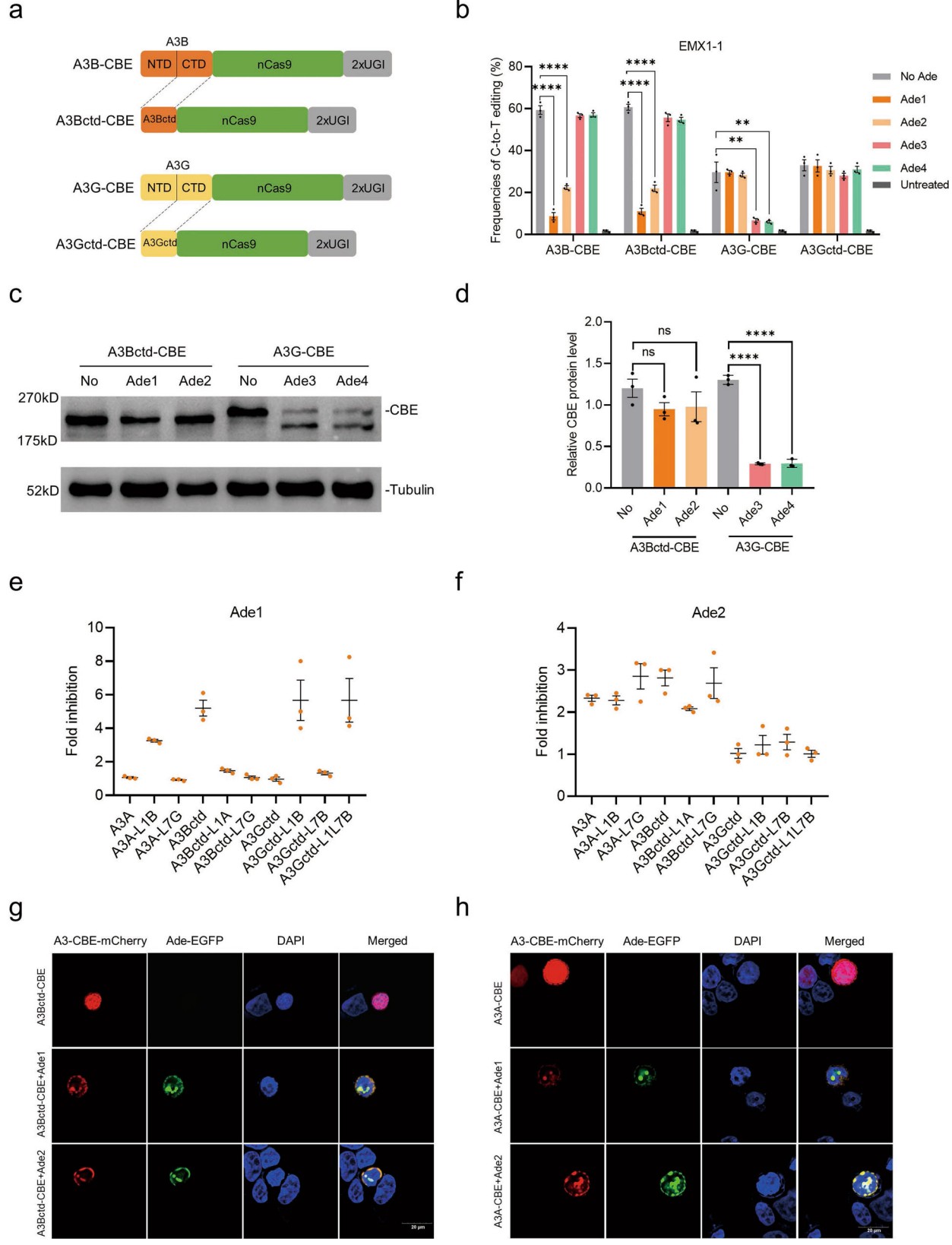

L7B compared with that of A3Gctd (1.32-fold vs. 0.96-fold), suggesting that the L7 region is also a prerequisite for the inhibitory action of Ade1 (Fig. 2e and Supplementary Fig. 3c). Furthermore, A3Gctd-L1L7B was also significantly inhibited compared with A3Gctd (5.66-fold vs. 0.96-fold) (Fig. 2e and Supplementary Fig. 3c). In contrast, Ade2 showed obvious

inhibition of all A3A- and A3Bctd-chimeras but exhibited no impact on any of the A3Gctd-chimeras, indicating that Ade2 may rely on regions other than L1 and L7 to co-exert inhibitory effects (Fig. 2f and Supplementary Fig. 3c). These A3 chimeras showed different editing efficiencies, editing windows and sequence context specificities, consistent with previous reports that L1

**Fig. 2 Inhibitory mechanism of Ades. a** Schematic representation of the A3B, A3Bctd, A3G and A3Gctd-CBE architectures. nCas9, D10A. **b** Base editing of A3B, A3Bctd, A3G and A3Gctd-CBE in the presence or absence of Ade1-Ade4 at the EMX1-1 site. Plasmids expressing CBE, sgRNA, and each Ade (1:1:1) were cotransfected into HEK293T cells. **c** Immunoblots of A3Bctd- and A3G-CBE in the presence or absence of the corresponding Ades. Tubulin was used as a loading control. **d** Quantification of the relative CBE protein contents normalized to tubulin in **c**. **e, f** Fold inhibition of base editing by Ade1 (**e**) and Ade2 (**f**) at the FANCF site. **g, h** Representative images of HEK293T cells expressing the A3Bctd-CBE-mCherry (**h**) and A3A-CBE-mCherry (**g**) constructs alone or in combination with Ade-EGFP constructs. The immunofluorescence microscopy experiment was repeated three times independently with similar results. Values and error bars reflect the mean ± s.e.m. and *n* = 3 biologically independent experiments. All *p* values were calculated by two-sided *t* tests. **\*\*p < 0.01, \*\*\*\*p < 0.0001.** Source data are provided as a Source Data file.

and L7 influence deaminase activities and intrinsic sequence preferences[30,31] (Fig. Supplementary Fig. 3d).

Viral ribonucleotide reductases (Ade1 and Ade2) were reported to inhibit APOBEC3 by relocating it from the nucleus to the cytosol[19]. To characterize how they affect base editors, we used an immunofluorescence microscopy approach to locate Ades and A3-CBEs in human cells after transfection. A3Bctd- or A3A-CBE alone showed obvious nuclear localization; however, they were also partially relocated to the perinuclear region by both Ade1 and Ade2 (Fig. 2g, h and Supplementary Fig. 4). Our results were different from those of a previous report showing that A3 deaminases were relocated from nuclear to cytoplasmic bodies[19], which may have been a result of the presence of a nuclear localization signal (NLS) in our CBE constructs. Both Ade1 and Ade2 relocated A3-CBEs from the nucleus to the perinuclear region, while they showed different inhibitory effects on A3-CBEs, suggesting that they might adopt other mechanism of inhibition.

Therefore, we tested that if Ade1 and Ade2 proteins directly inhibit deaminase activity of A3 deaminases by an in vitro deaminase activity assay. As a result, Ade1 potently inhibited A3Bctd and Ade2 moderately inhibited both A3A and A3Bctd, consistent with the results in human cells (Supplementary Fig. 5). Interestingly, the Ade1 slightly inhibited A3A in vitro, while no evident inhibition of A3A-CBE was observed in human cells (Supplementary Fig. 5). We suspected that there is a certain threshold of inhibition intensity to achieve effective inhibition in vivo, which also explains why some inhibitors (Ade5-Ade7) that are effective in vitro while defunct in vivo.

Taken together, these results demonstrated that Ade1 and Ade2 affect base editors by inhibiting deaminase activity and relocation. Specifically, Ade1 interacts with the L1 and L7 regions, and both Ade1 and Ade2 relocate A3-CBEs from the nucleus to the perinuclear region. Ade3 and Ade4 inhibit A3G-CBE by degrading the whole A3G deaminase.

## Expanding the application of Ades to CGBE, A&CBE and A&CGBE.
It is well known that rat APOBEC1 (rA1)-CBE is the most widely used CBE system, followed by the A3A-, eAID-, eCDA1-, A3G- and other CBE systems[32–34]. However, non-A3-CBEs cannot be inhibited by the Ades used in this study (Supplementary Fig. 6). We intended to replace rA1-CBE with A3B- and A3Bctd-CBE since they have excellent C-to-T editing efficiency and can be greatly inhibited by Ade1. Therefore, we compared the base editing characteristics of the rA1-, A3B- and A3Bctd-CBE systems at seven target sites (Supplementary Fig. 7a–c). Notably, the average efficiency of A3Bctd-CBE (56.7 ± 2.1%) was comparable to that of classical rA1-CBE (56.4 ± 1.4%), and A3B-CBE (51.8 ± 2.6%) was slightly less efficient (Supplementary Fig. 7d). In addition, A3Bctd-CBE and A3B-CBE showed a similar but slightly broader editing window than rA1-CBE (Supplementary Fig. 7e). Importantly, Ade1 and Ade2 exhibited strong and moderate inhibition of A3Bctd-CBE at all seven tested sites, respectively (Supplementary Fig. 7f). These

data demonstrated that A3Bctd-CBE can replace conventional rA1-CBE for efficient base editing.

Other BE systems, except for CBE, including CGBE, which induces C-to-G transversion, and A&CBE, which simultaneously induces C-to-T and A-to-G conversions, were developed based on the rA1 deaminase[35–40]. Here, we constructed nine new A3A variant- and A3Bctd variant-CGBEs by introducing a series of mutations into the A3A-nCas9 and A3Bctd-nCas9 architectures[33,35,41,42] (Fig. 3a). As a result, the C-to-G efficiencies of the newly designed A3A-N57G, A3A-Y130F, A3Bctd-R211A and A3Bctd-R211K systems were comparable to those of the classical rA1-R33A system when tested at four sites (Supplementary Fig. 8). Each CGBE performed differently in specific sites, as reported in a previous study[43]. Moreover, the four best-performing A3A and A3Bctd variant-CGBEs were significantly inhibited by Ade1 and Ade2 (Fig. 3b, c). We observed that engineering deaminase is an efficient method to improve C-to-G editing efficiency, consistent with a previous report of rA1-R33A[35]. Although the mechanism is not clear, we reasoned that narrowing the deamination window of the deaminase might increase the C-to-G editing outcome.

Subsequently, we constructed an A3Bctd-A&CBE using the A3Bctd-8e-nCas9-2xUGI architecture, in which 8e is a state-of-the-art ABE[44] (Fig. 3d). Both C-to-T and A-to-G editing was efficient at two target sites (Fig. 3e, f). Interestingly, Ade1 inhibited both cytosine and adenine base conversions produced by A&CBE, but C-to-T editing was inhibited to a larger extent (Fig. 3e, f). Furthermore, we designed a novel A&CGBE using the A3Bctd-R211A-8e-nCas9 architecture (Fig. 3g), which induced both C-to-G and A-to-G editing efficiently at the RNF2 site but not at ABE site7 (Fig. 3h, i). Notably, Ade1 significantly suppressed C-to-G editing and only slightly attenuated A-to-G editing at the RNF2 site (Fig. 3h). Taken together, these results suggested that various BE systems comprising A3A and A3Bctd enable various types of base editing and can be regulated by Ade-off switches.

## Expanded inhibitory ranges of Ades to rA1-CBE and ABE8e.
The A3A- and A3Bctd-BEs significantly expanded the inhibitory range, while the classical rA1-CBE and ABE system still could not be inhibited by current Ades (Supplementary Fig. 6). In addition, the method of exchanging L1 and L7 in A3 deaminases to achieve inhibition is not practical for rA1 or ABE due to their significant sequence differences. However, we noticed that Ade3 and Ade4 use a different inhibitory mechanism, degrading the A3G deaminase. It has been reported that the N-terminal domain of A3G (A3Gntd) binds to Ade3, and the smallest fragment of A3G containing only amino acids 105–245 can trigger Ade3-induced degradation[45,46]. On the basis of these considerations, we hypothesized that fusing A3Gntd to other BE systems can extend the Ade3 inhibitory ranges. Therefore, we first constructed a series of A3Gntd-rA1-CBEs by connecting A3Gntd segments of various lengths (1–196, 1–245, 105–196 and 105–245) to the N-terminus of rA1-CBE (Fig. 4a). As a result, Ade3 and Ade4 showed no obvious inhibition of all four A3Gntd-rA1-CBEs, while full-

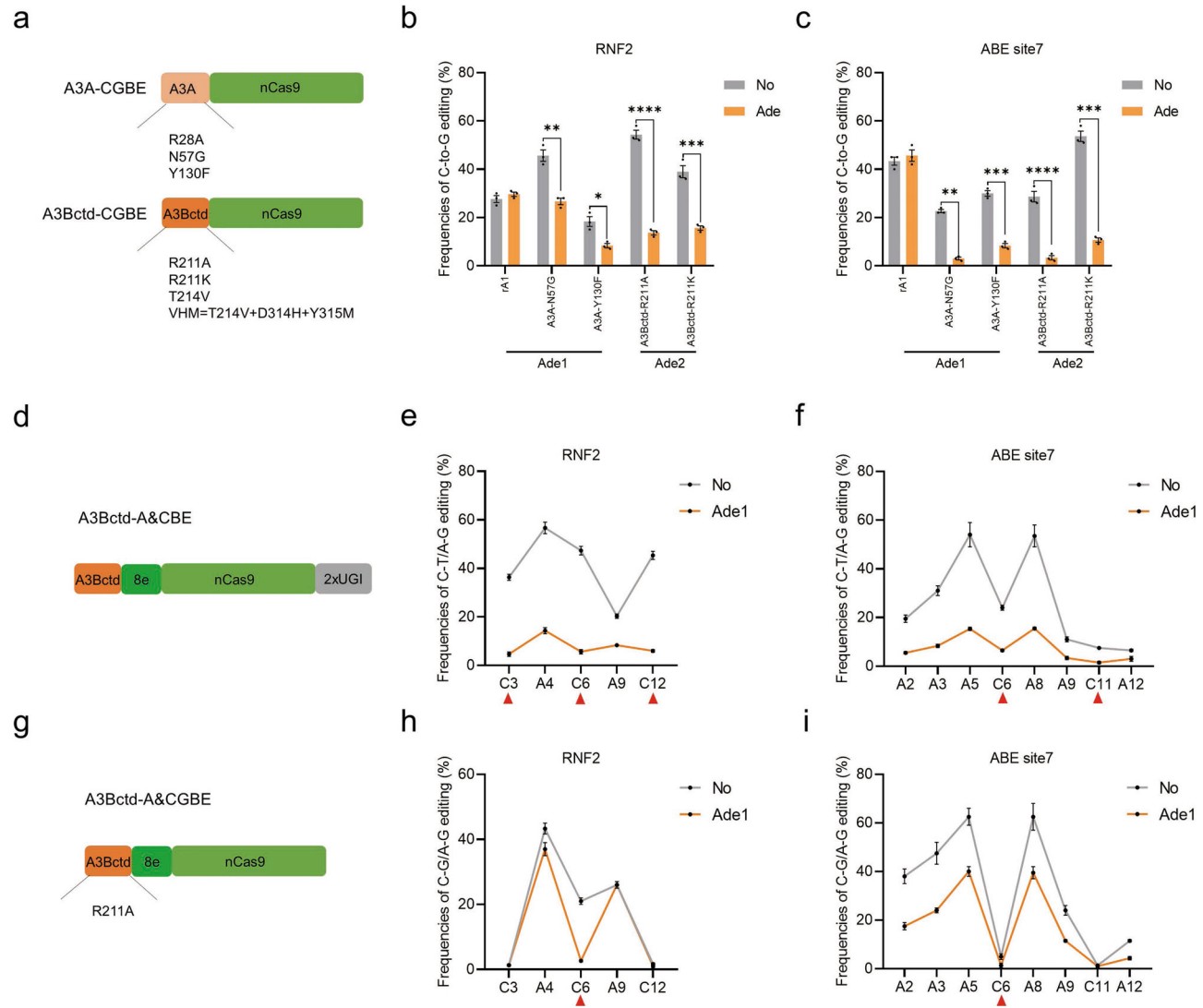

**Fig. 3 Expanding the application of Ades to CGBE, A&CBE and A&CGBE. a** Schematic representation of engineered A3A variant- and A3Bctd variant-CGBEs. nCas9, D10A. **b** Bar plots showing the base editing frequencies induced by rA1 variants, A3A variants-, and A3Bctd variants-CGBEs at the RNF2 site. Target C, red. **c** C-to-G editing frequencies of four representative CGBEs in the presence or absence of Ade at the RNF2 site. Plasmids expressing CGBE, sgRNA, and Ade (1:1:1) were cotransfected into HEK293T cells. **d** Schematic representation of engineered A3Bctd-A&CBE. nCas9, D10A. **e**, **f** Frequencies of single C-to-T and A-to-G conversions using A3Bctd-A&CBE in the presence or absence of Ade1 at RNF2 (**e**) and ABE site7 (**f**). Target C, red triangle. **g** Schematic representation of engineered A3Bctd-A&CGBE. nCas9, D10A. **h**, **i** Frequencies of single C-to-G and A-to-G conversions using A3Bctd-A&CGBE in the presence or absence of Ade1 at RNF2 (**h**) and ABE site7 (**i**). Target C, red triangle. Values and error bars reflect the mean ± s.e.m. and $n = 3$ biologically independent experiments. All $p$ values were calculated by two-sided t tests. *$p < 0.05$, **$p < 0.01$, ***$p < 0.001$, ****$p < 0.0001$. Source data are provided as a Source Data file.

length A3G-CBE was significantly inhibited at the two tested sites (Fig. 4b, c). We speculated that the divergence may lie in the difference between the in vitro and in vivo experimental environments and the influence of the architecture of cytidine deaminase-nCas9 fusion. Subsequently, we inactivated full-length A3G (dead A3G, dA3G) by introducing an E259A mutation to abolish its catalytic activity and constructed dA3G-rA1-CBE[47] (Fig. 4a). Notably, the C-to-T editing frequencies of dA3G-rA1-CBE were moderately decreased in the presence of Ade3 and Ade4, compared with those of the control group (Fig. 4b, c). Encouraged by this result, we fused dA3G to the N-terminus of ABE8e (Fig. 4d). Notably, dA3G-ABE8e was significantly inhibited by both Ade3 and Ade4, while ABE8e was not inhibited at the two target sites (Fig. 4d). These results indicated that the inhibitory ranges of Ades were further expanded to rA1-CBE, particularly ABE8e, with an engineered dA3G domain.

**Inhibition of Cas9-dependent and Cas9-independent off-target editing by Ades.** The recognition of cognate protospacer sequences of CBEs can cause severe Cas9-dependent OT effects[3,5]. In addition, it has been reported that CBEs can potentially create stochastic Cas9-independent OT edits on the genome and transcriptome[6–9]. The potential safety risk caused by DNA off-target edits is higher than that caused by RNA off-target edits, so we focused on the DNA OT editing of CBEs. First, a classical EMX1-1 target site was selected to evaluate on-target editing and Cas9-dependent OT editing (Fig. 5a, b). A recently reported sensitive and cost-effective orthogonal R-loop assay was used to assess the Cas9-independent OT effect on the SaCas9-induced R-loop region, including the Sa site5, site6, site14 and site17, which had been detected in previous reports[4,27,48] (Fig. 5c). Conventional AcrIIC1 (no inhibition of SpCas9, as a negative control), AcrIIA5 (strong inhibition of SpCas9) and

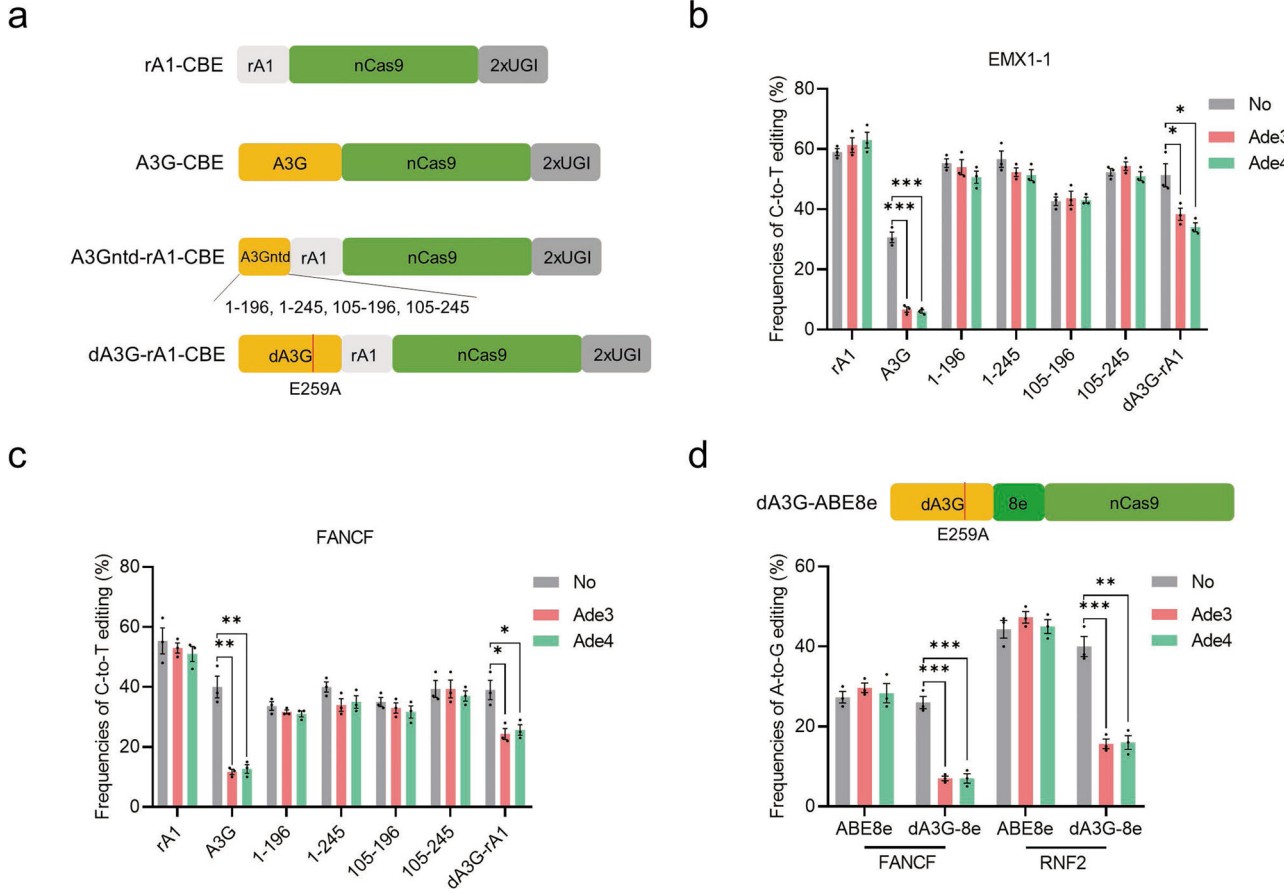

**Fig. 4 Expanding the application of Ades to rA1-CBE and ABE8e. a** Schematic representation of rA1-, A3G-, A3Gntd-rA1- and dA3G-rA1-CBEs. nCas9, D10A. **b**, **c** C-to-T editing of seven CBEs in the presence or absence of Ade3 and Ade4 at the EMX1-1 site (**b**) and FNACF site (**c**). Plasmids expressing CBE, sgRNA, and each Ade (1:1:1) were cotransfected into HEK293T cells. **d** Top: Schematic representation of dA3G-ABE8e. Bottom: A-to-G editing of ABE8e and dA3G-ABE8e in the presence or absence of Ade3 and Ade4 at the FANCF and RNF2 sites. Values and error bars reflect the mean ± s.e.m. and $n = 3$ biologically independent experiments. All $p$ values were calculated by two-sided $t$ tests. *$p < 0.05$, **$p < 0.01$, ***$p < 0.001$. Source data are provided as a Source Data file.

Ade1 and Ade2 were used for parallel comparisons[49]. As expected, AcrIIA5 efficiently inhibited the on-target editing of both A3A-CBE (3.0-fold) and A3Bctd-CBE (4.8-fold) (Fig. 5a). Ade1 inhibited A3Bctd-CBE (5.0-fold), and Ade2 inhibited both A3A-CBE (1.3-fold) and A3Bctd-CBE (2.3-fold) (Fig. 5a). However, AcrIIA5 significantly reduced Cas9-dependent OT editing but maintained a high level of Cas9-independent OT editing, consistent with the inhibition mechanism by which AcrIIA5 interacts with the Cas9 domain (Fig. 5b, c). Notably, Ade1 almost completely suppressed both Cas9-dependent and Cas9-independent OT editing of A3Bctd-CBE, and Ade2 significantly reduced the OT editing of both A3A- and A3Bctd-CBE (Fig. 5b, c). Furthermore, both Ade1 and Ade2 exhibited stronger inhibition of Cas9-independent OT activities than AcrIIA5 (Fig. 5c). Similar results were observed at the HEK site4 (Supplementary Fig. 9). Ade1 also showed a slight decrease in OT editing of A3A-CBE without compromising on-target editing (Fig. 5b, c). This might be explained by our observations showing that Ade1 slightly inhibited A3A deaminase activity and relocates A3A-CBE from the nucleus to perinuclear bodies. A3Gctd-CBE is of high specificity naturally in both Cas9-dependent and Cas9-independent OT sites, as reported in previous reports[31,50,51] (Supplementary Fig. 10). Overall, these results suggested that Ades can suppress both Cas9-dependent and Cas9-independent OT editing, holding the potential of being safer off switches of CBEs than current Acrs.

**Cell-specific CBE editing by microRNA-responsive Ade switches.** To render CBE-based gene modifications and therapies precise and safe, strategies that confine the activity of a CBE to specific cells and tissues are highly desired. In recent studies, Acr proteins have been used to achieve cell-specific CRISPR-Cas9 editing, activation and even tissue-restricted genome editing in vivo by microRNA-dependent expression of Acr proteins[52–54]. Given that Ades can inhibit both Cas9-dependent and Cas9-independent OT activities, they make safer switches for CBEs than traditional Acrs. Therefore, we tried to develop a cell type-specific Cas-ON switch based on miRNA-regulated expression of Ade proteins. We inserted binding sites of miR-122 (miR-122BS), which are abundant specifically in liver cells, into the 3′UTR of Ade transgenes (Fig. 6a). Coexpressing these with A3Bctd-CBE and sgRNAs resulted in Ade knockdown and released CBE activity exclusively in hepatocytes (such as in Huh7 cells), while CBE was inhibited in miR-122-free cells (such as HEK293T cells) (Fig. 6a). In HEK293T cells, Ade1 robustly inhibited C-to-T editing by A3Bctd-CBE at two tested sites (Fig. 6b, c). The presence or absence of miR-122BS showed no significant effect on editing inhibition in the miR-122-free cells (Fig. 6b, c). The editing frequency was decreased in Huh7 cells compared with that in HEK293T cells due to the lower transfection efficiencies in Huh7 cells, as previously reported[52,54]. Ade1 also prevented editing in Huh7 cells when expressed by constructs that lacked miR-122BS. In contrast, Ade1 plasmids incorporating miR-122BS

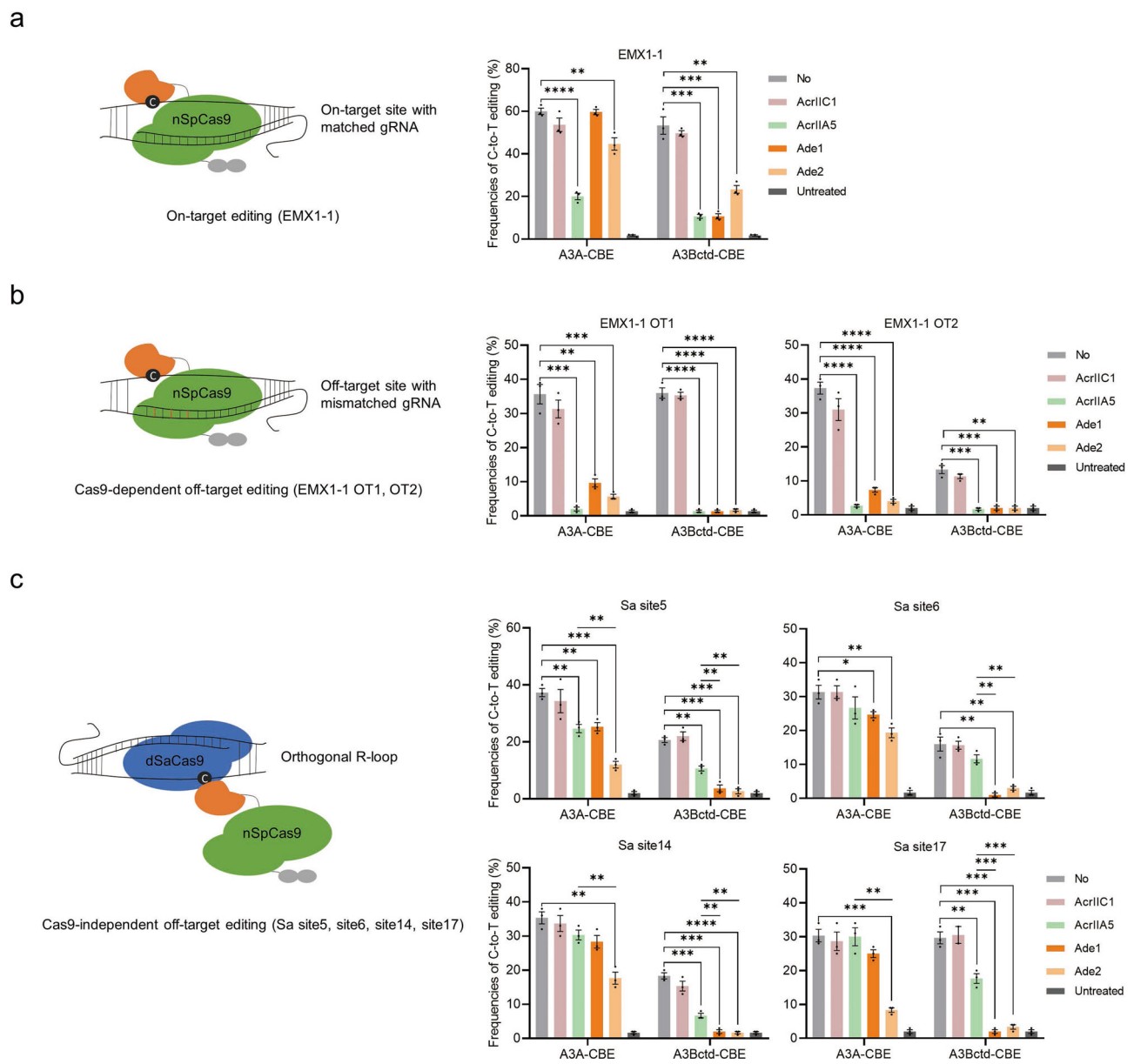

**Fig. 5 Evaluation of Cas9-dependent and Cas9-independent OT editing.** The principle of the assay is illustrated on the left, and the results are shown as bar graphs on the right. **a** On-target editing frequencies of A3A and A3Bctd-CBE in the presence or absence of AcrIIC1 (as a negative control), AcrIIA5, Ade1 and Ade2 at the EMX1-1 site. **b** Cas9-dependent OT editing frequencies of A3A and A3Bctd-CBE in the presence or absence of AcrIIC1 (the negative control), AcrIIA5, Ade1 and Ade2 at the EMX1-1 OT1 and OT2 sites. **c** Cas9-independent OT editing frequencies of A3A and A3Bctd-CBE in the presence or absence of AcrIIC1 (the negative control), AcrIIA5, Ade1 and Ade2 at Sa site5, site6, site14 and site17. The dSaCas9 together with a gRNA was used to induce a stable ssDNA region (orthogonal R loop) at a specific locus, thus artificially magnifying Cas9-independent deamination. Values and error bars reflect the mean ± s.e.m. and $n = 3$ biologically independent experiments. All $p$ values were calculated by two-sided t tests. $*p < 0.05$, $**p < 0.01$, $***p < 0.001$, $****p < 0.0001$. Source data are provided as a Source Data file.

in the 3′UTR failed to inhibit CBE editing in Huh7 cells, as indicated by editing efficiencies that were similar to the no-Ade control (Fig. 6b, c). These results illustrated that the developed CBE-ON switch can confine CBE activity to the selected cell types, facilitating safe and precise CBE-based gene modifications and therapies.

## Discussion

In this study, we identified four virus-derived Ade proteins that can be utilized to efficiently inhibit CBE editing in vivo. By rationally engineering deaminases, we successfully utilized Ades with various types of BEs, including CGBE, A&CBE, A&CGBE, rA1-CBE and even ABE8e. We found that Ades were safer off switches for CBEs than conventional Acrs because they directly inhibit the activities of cytidine deaminase, the effector domain in CBEs. In addition, the designed CBE-ON switch provided a safe and precise regulator for CBE applications.

The APOBEC3 family of cytosine deaminases is an integral part of the antiviral innate immune response, which inhibits virus replication through deamination-dependent and deamination-independent activities[19]. Viruses have evolved mechanisms to counteract these enzymes, such as the well-studied Ade3 (HIV-1 Vif)-mediated formation of an ubiquitin ligase to degrade virus-restrictive A3G enzymes[47,55]. Since Vif simply reduces the amounts of A3G-CBE in cells, it might not be a good candidate

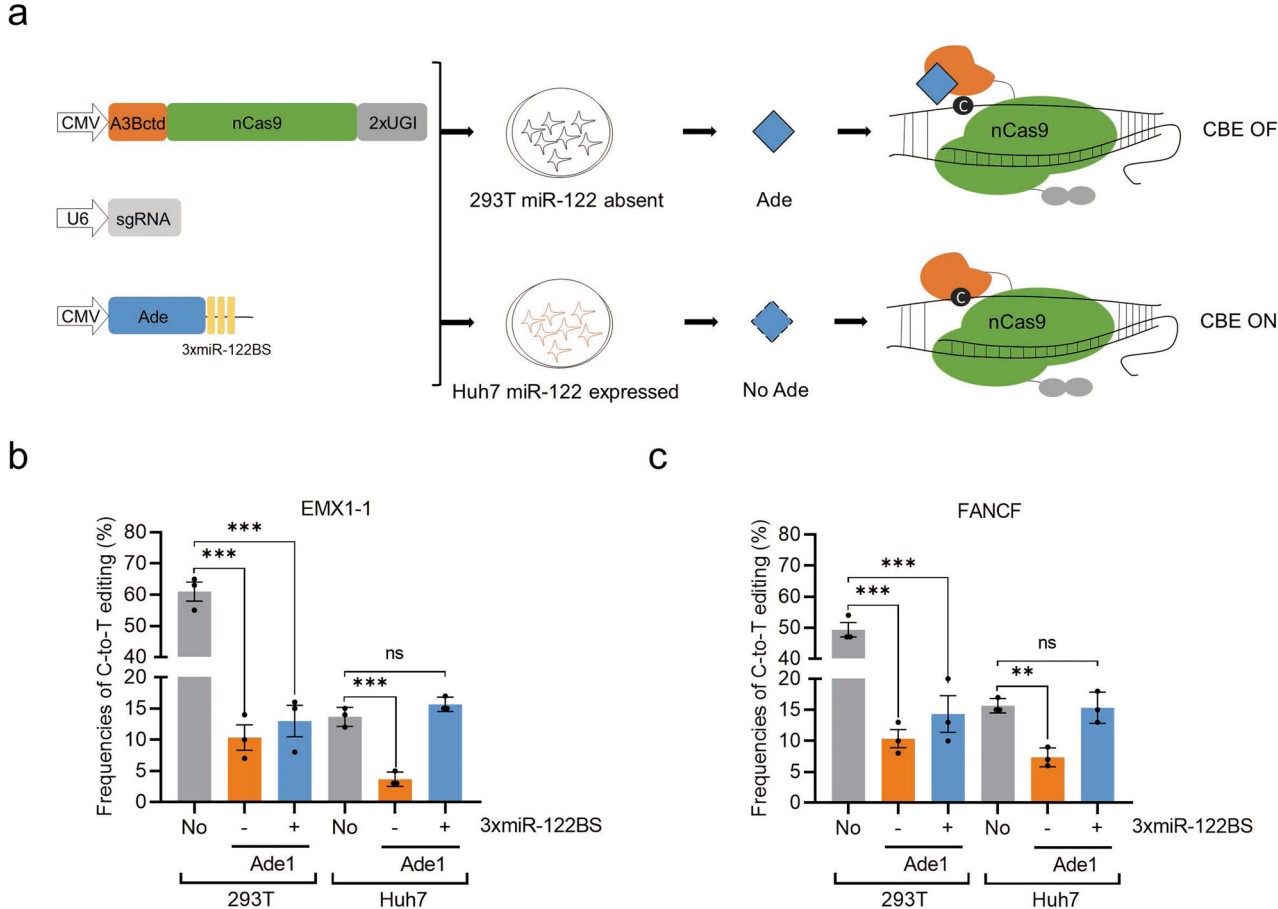

**Fig. 6 Hepatocyte-specific base editing by A3Bctd-CBE in cultured cells. a** Schematic of the CBE-ON switch. The miR-122 binding sites were inserted into the 3'UTR of an Ade-encoding transgene. Upon delivery, Ade expression is selectively knocked down within the target cells, permitting CBE activity. In OFF-target cells lacking the miR-122 signature, the Ade protein is translated and inhibits A3Bctd-CBE. **b, c** HEK293T and Huh7 cells were transfected with plasmids encoding A3Bctd-CBE, sgRNA and Ade1 with or without 3xmiR-122BS at the EMX1-1 site (**b**) and FANCF site (**c**). Values and error bars reflect the mean ± s.e.m. and $n = 3$ biologically independent experiments. All $p$ values were calculated by two-sided t tests. **$p < 0.01$, ***$p < 0.001$. Source data are provided as a Source Data file.

for specifically reducing off-target deamination while maintaining the efficacy of on-target deamination. However, we demonstrated that both Ade3 and Ade4 can inhibit A3G-CBE activity, holding the potential to as off switches to regulate CBE in specific applications. In addition, virus ribonucleotide reductases (RNRs) were found to mediate the inhibition of A3 deaminases in recent years[17,19]. The large subunits of the viral RNRs, including Ade1 and Ade2, bind to A3A and A3B and relocate them from the nucleus to cytoplasmic bodies[19]. Ade1 (EBV-BORF2) was reported to directly inhibit A3B catalytic activity in an in vitro deaminase activity assay[17]. We further demonstrated that both Ade1 and Ade2 directly inhibit A3A and A3Bctd deaminase activity and that Ade1 specifically interacts with the L1 and L7 regions of A3Bctd. Recently, the cryo-EM structure of BORF2-A3Bctd showed that L1 and L7 of A3Bctd have multiple contact points with BORF2, consistent with our results[56].

Ade2 is a weak inhibitor of A3A-BE that maintains high on-target efficiency but significantly reduces Cas9-dependent and Cas9-independent OT editing. In addition, we also noticed that Ade2 moderately narrowed the C-to-T editing window of A3A-CBE at the tested sites (Supplementary Fig. 11). A similar phenomenon was observed in a recent study of Cas9 inhibitors[57]. It has been reported that coupling Cas9 to artificially weakened Acr proteins to fine-tune its activity towards selected sites can enhance CRISPR-Cas9 target specificity[58]. Based on these data,

we believe that Ade2, as a naturally weak inhibitor of A3A-CBE, has the potential to be artificially engineered using a similar approach to further improve the specificity of A3A-CBE.

We noticed that Ades did not completely negate all base editing activities of A3-CBEs at some target sites, especially in engineered non-A3-CBEs. This may be a bottleneck for the application of Ades when CBE activity needs to be completely shut down, especially for CBE-based therapeutic applications. In addition, we found that cotransfection of both AcrIIA5 and Ade1/Ade2 further enhanced the suppression of base editors (Supplementary Fig. 12). Moreover, computational design of anti-CRISPR proteins has been demonstrated to improve inhibition potency[59], and it can be used to enhance Ades in the future.

Virus-derived APOBEC antagonists comprise a huge treasure trove. Further exploration is necessary to find efficient and specific Ades in addition to those tested in this study. In addition to virus-derived Ades, it has been reported that single-stranded binding proteins, small-molecule inhibitors and chemically modified oligonucleotides can suppress the cytosine deaminase activity of APOBEC deaminases[60-65]. These potential antagonists of deaminases merit further exploration to develop new inhibitors of CBEs in the future.

In summary, we developed a series of Ades derived from viruses that can efficiently inhibit C-to-T editing of CBEs. Moreover, we expanded the Ade target ranges to various types of

BEs by rationally engineering deaminases. Importantly, these Ades can dramatically decrease both Cas9-dependent OT activity and Cas9-independent OT activity and can be used to generate a cell type-specific CBE-ON switch based on a microRNA-responsive Ade vector. Thus, these Ades, with efficient inhibition of CBEs, are promising tools for future gene modification and therapeutic applications.

## Methods

**Plasmid construction**. The rA1-CBE, eCDA1-CBE, dSaCas9 and ABE8e plasmids were obtained from Addgene (#112093, #122608, #138162 and #138489). The DNA fragments of Ade1-Ade7 were synthesized and cloned into the pcDNA3.1 vector (GenScript). The DNA fragments of A3A, A3B, A3Bctd, A3Gctd, A3Gntd, dA3G and miR-122BS were synthesized and cloned into the corresponding vectors (GenScript). Plasmid site-directed mutagenesis was performed using a Fast Site-Directed Mutagenesis Kit (Tiangen). The sequences of plasmids are listed in the Supplementary sequence.

**Cell culture and transfection**. HEK293T (ATCC) and Huh7 (ATCC) cell lines were cultured in Dulbecco's modified Eagle's medium (DMEM) supplemented with 10% fetal bovine serum (HyClone) and incubated at 37 °C in an atmosphere of 5% $CO_2$. The cells were seeded in 24-well plates and transfected using Hieff Trans™ Liposomal Transfection Reagent (Yeasen) according to the manufacturer's instructions. Puromycin (Meilunbio) was added at a final concentration of 3 µg/mL to enrich the positively transfected cells 24 h after transfection. After 72 h, the cells were collected and used for genotyping by EditR[66]. All target sites and primers used for genotyping are listed in Supplementary Table 1 and Supplementary Table 2.

**Western blotting**. Western blotting analyses were performed as described previously[67]. Briefly, HEK293T cells were grown in 6-well plates and transfected with 1000 ng of A3-CBE plasmids (A3A-, A3B-, A3Bctd-, A3G and A3Gctd-CBEs) and 1000 ng of EMX1-1 sgRNA plasmid with or without Ade plasmids (Ade1–4). Seventy-two hours later, the transfected cells were lysed in RIPA buffer supplemented with a protease inhibitor cocktail (Roche, Basel, Switzerland). The whole-cell lysate was used for immunoblotting. An antibody against Cas9 (1:1500; ab204448, Abcam) was used as a primary antibody, while an anti-tubulin antibody (1:2000; 10094-1-AP, Wuhan Sanying) was used as the loading control.

**Immunofluorescence microscopy**. HEK293T cells were grown in 6-well plates and transfected with 1500 ng of A3-CBE-mCherry plasmid alone, 1500 ng of Ade-EGFP plasmid alone or both 1500 ng of A3-CBE-mCherry and of Ade-EGFP. After 48 h, the cells were fixed in 4% formaldehyde, permeabilized in 0.2% Triton X-100 in PBS for 10 min, and washed three times for 5 min each time in PBS. Cells were then counterstained with 1 µg/ml DAPI for 5 min and rinsed twice for 5 min each time in PBS and once in sterile water. Coverslips were mounted on precleaned slides using 20 µl of mounting medium. The slides were imaged with an Olympus FV1000 microscope instrument.

**Protein purification and in vitro deaminase activity assays**. Codon-optimized A3A, A3Bctd, Ade1 and Ade2 were synthesized and ligated into pSmartI vector for *E. coli* expression and protein purification, which were carried out by Gene Universal. In vitro deamination activity assay was performed similarly to the published method[17]. In brief, a fluorescent oligo substrate (5′-ATT ATT ATT ATT CAA ATG GAT TTA TTT ATT TAT TTA TTT ATTT-6-FAM-3′) was synthesized (Sangon). In vitro deamination experiment mixture contained 1 µl recombinant A3 protein, 1 µl 3.3 µM oligo, 0.5 µl UDG (New England Biolabs), and 7.5 µl modified 2-hydroxyethyl disulfide (HED) buffer (20 mM HEPES, 50 mM NaCl, 0.1 mM EDTA, 0.1 mg ml$^{-1}$ BSA, pH 7.4). Then the mixture was incubated at 37 °C for 30 min; 1 µl of 1.1 M NaOH was then added and heated to 98 °C for 5 min to cleave the DNA at abasic sites. The reaction was then mixed with 11 µl 2×formamide buffer and run on a 15% TBE-urea PAGE gel. Separated DNA fragments were visualized on Azure c600 scanner on fluorescence mode (Azure Biosystems). A3 deaminase activity was quantified on ImageJ by dividing product band intensity by the sum of product and substrate band intensities

**Off-target assay**. The Cas9-dependent off-target sites for EMX1-1 and HEK site4 were widely used in previous reports[3,31]. An orthogonal R-loop assay to evaluate Cas9-independent off-targets was performed as previously described[4,51]. Briefly, to check off-target editing at Sa site5, plasmids expressing dSaCas9 (200 ng) and an sgRNA targeting Sa site5 (200 ng) were cotransfected into HEK293T cells with plasmids expressing CBE (200 ng), an sgRNA targeting EMX1-1 (200 ng) and Ade/Acr (200 ng). Cells were treated with puromycin, and editing at both Sa site5 and EMX1-1 was detected by Sanger sequencing as described in the previous section. All primers for the off-target assay are listed in Supplementary Table 2.

**Statistical analysis**. All data are expressed as the mean ± s.e.m. of at least three individual determinations for all experiments. The data were analyzed by two-sided t test via GraphPad Prism software 8.0.1. A probability value smaller than 0.05 ($p < 0.05$) was considered to be statistically significant. $*p < 0.05$, $**p < 0.01$, $***p < 0.001$, $****p < 0.0001$.

**Reporting summary**. Further information on research design is available in the Nature Research Reporting Summary linked to this article.

## Data availability

All data generated or analyzed during this study are included in this published article and its supplementary files. Source data are provided with this paper.

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

## Acknowledgements
This study was financially supported by the National Key Research and Development Program of China Stem Cell and Translational Research (2017YFA0105101, Z.J.L.) and The National Natural Science Foundation of China (Nos.32170543 and 31970574, Z.J.L.).

## Author contributions
Z.Q.L., L.X.L. and Z.J.L. conceived and designed the experiments. Z.Q.L. and S.Y.C. performed the experiments and analyzed the data. Z.Q.L. and Z.J.L. wrote the paper. All authors read and approved the final manuscript.

## Competing interests
The authors declare no competing interests.
