## [Peer Review File · Nature Communications]

Reviewers' Comments:

Reviewer #1:

Remarks to the Author:

The manuscript by Liu et al. report on a series of anti-deaminases (Ades) derived from viruses, and subsequently identified four Ades that efficiently inhibit the activity of APOBEC3 deaminases, which is capable of inhibiting cytosine base editors (CBEs). Based on a structure analysis of A3 deaminases, they noticed that various Ades have different deaminase inhibitory mechanisms. Importantly, the authors demonstrate that the use of Ades decreases both Cas9-dependent DNA off-target activity and Cas9-independent DNA off-target activity of CBEs. Furthermore, the Ades could also inhibit other base editors such as CGBE, A&CBE, A&CGBE, rA1-CBE and even an adenine base editor (ABE8e) through additional engineering of the deaminase domain. They also explored a cell type-specific CBE-ON switch based on a microRNA-responsive Ade vector.

To the best of our knowledge, this work is the first report discussing anti-deaminases for efficiently inhibiting base editing activities, which has significant novelty to be published in Nature Communications. However, I believe that an on-/off-target analysis of the anti-deaminases suggests that these inhibitors are globally reducing base editing activity and not solely affecting just off-target editing. Because the discovery of these inhibitors is itself novel, I recommend the authors re-write the manuscript to focus on the discovery and analysis of these inhibitors rather than on their propensity to affect off-target base editing activities. I recommend publication of this manuscript in Nature Communications but the authors should re-focus the paper on the anti-deaminases rather than on their ability to affect off-target editing.

Major comments:

1. The authors stated that the Ades can suppress both Cas9-dependent and -independent off-target DNA editing. However, the authors only selected one site (EMX1-1) with only two Cas9-dependent and two Cas9-independent off-target sites to evaluate in this manuscript, which is insufficient to support their strong conclusions (Fig. 3). The authors should evaluate additional sites to thoroughly evaluate this effect. Alternatively, the authors can re-focus the manuscript to not heavily rely on the ability of these inhibitors to change on- to off-target editing activities.
2. There is a lack of no-treatment controls presented in all figures. I wonder whether certain base mutagenesis events in some figures are a result of "background mutations" in the cell; therefore, I strongly recommended the authors include a control group for each figure.
3. It seems that the Ades inhibited the activity of base editors, but did not completely negate all base editing activities (the base editors can still induce both on- and off-target edits at a low level even when treated with Ades), which may be a bottleneck for various application of Ades, especially CBE-based therapeutic applications. Thus, I recommend the authors discuss whether the editing efficiencies following treatment with Ades is safe or precise enough in the revised manuscript. Furthermore, I recommend the authors explore the combination of using both AcrIIA5 and Ades and evaluate whether this would enhance the suppression of base editors.
4. The authors declared that "The inhibitory ranges of Ade were further expanded to CGBE, A&CBE, A&CGBE, rA1-CBE and even ABE8e by rationally engineering the deaminases domain." in the Abstract. However, I believe that this statement is an over-exaggeration as the Ades only modestly decrease the activities of other base editors according to the data shown in the manuscript. Secondly, certain deaminase activities are not affected by Ades at particular target sites (such as ABE8e activity in Fig. 4H). Therefore, I recommend the authors soften their conclusions.
5. The results in Fig. 4B are very interesting and exciting, which showed that variants of rA1, A3A, and A3Bctd significantly altered the purity of CGBEs. However, the authors only evaluated two sites, which is insufficient to draw the conclusion that A3A-N57G, A3Bctd-R211A, and A3Bctd-R211K are the best candidates for generating C-to-G base editing (for example, C-to-G efficiency of A3Bctd-R211A is lower than A3Bctd and A3Bctd-R211K at ABE site7 in Fig. S5, and C-to-G efficiency of A3Bctd-R211K is lower than A3Bctd-R211A and A3Bctd-VHM). Thus, I recommend the authors evaluate additional sites and also test the editing effects of using A3A-CBE ("wild-type" version) in Fig. 4B. Moreover, I am confused as to why most of the variants engineered for other purposes (such as A3A-N57G [PMID: 30059493] is engineered for preferentially editing "TC" motifs, and A3Bctd-VHM [PMID:32721385] was engineered for reducing the Cas9-independent off-target effects) showed significantly increased C-to-G editing. I recommend the authors discuss their hypothesis on how these mutations affect C-to-G editing in the manuscript.

6. I recommend the authors highlight the species from which the anti-deaminases originated in the manuscript. Furthermore, to measure the cross-species activity of the Ades, I recommend the authors test whether certain Ades could inhibit the activities of deaminases from other species.
7. It seems that the ratio of Ade to CBE at 3:1 is not a maximal concentration for inhibition (Fig. 1E and Fig. 1F). I wonder if the author have tested higher doses of Ade1 and Ade2 for inhibition, and if so, please note this in the manuscript or methods?

Minor comments:

Line 32: "Cytosine base editors (CBEs), comprising cytidine deaminases fused to the N-terminus of Cas9 nickase (nCas9)" This sentence is inaccurate. Although the original version of CBEs, such as BE3, BE4 used cytidine deaminases fused to the N-terminus of Cas9 nickase, there are many newer variants with CBEs fused to other regions. I recommend the authors revise this sentence.

Line 34: I recommend the authors revise the sentence "However, the overexpression of cytidine deaminases leads to both Cas9-dependent off-target activities and Cas9-independent off-target activities...." since only the "Cas9-independent off-target activities" are induced by "overexpression of cytidine deaminases".

Line 40: For the sentence "Moreover, both Cas9-dependent off-target activity and Cas9-independent off-target activity of CBEs dramatically decreased at the presence of Ades, while the latter cannot be inhibited by the anti-CRISPRs (Acrs)", I believe the data does not reflect this phenomenon in the manuscript. According to Fig 3C, the AcrIIA5 (green column) can significantly decrease Cas9-independent off-target editing. I recommend the authors perform a statistical significance test comparing the AcrIIA5 and Ade1/2 to demonstrate the superiority of Ades for reducing Cas9-independent off-target editing.

Line 97: The authors mentioned "BE4max" so I recommend the authors cite the relevant manuscript (PMID: 29813047).

Line 102: Please check if "(Fig. 3C and 3D)." should be "(Fig. 1C and 1D)."

Line 127: "Ade1only" should be "Ade1 only".

Line 148, "A3-chimeras" should be "A3-chimeras".

Line 162: The "Sa site" should be detailed clarified, such as "SaCas9-induced R-loop region" or others.

Lines 181-184: The sentence "However, there is no natural inhibitor available for them, they cannot be inhibited by the Ades ... due to the low sequence homology with A3 deaminases". The use of competing pronouns makes this sentence extremely difficult to comprehend. I recommend the authors revise and clear up this sentence.

Line 301: The fonts of all "#" should be identical.

Lines 326 and Line 327: "200ng" should be "200 ng".

Lines 352-481: The format of all references should be identical. Some references lack a volume number and/or page number (such as Ref. 5, 13, 14, 16 etc.). Some references lack or use an incorrect journal name (such as Ref. 50, 55).

Reviewer #2:

Remarks to the Author:

This manuscript reports inhibition of cytosine deaminases coupled with the Cas9 nickase by using inhibitors derived from viruses. Deamination of off-target nucleotides is a problem that must be overcome in order to develop base-editing as a therapeutic method. The authors show that EBV-BORF2 (Ade1) and KSHV-ORF61 (Ade2) have a potential to reduce the off-target deamination caused by APOBEC3A (A3A) or APOBEC3B (A3B) fused Cas9 base-editors.

The authors further experimented chimera-A3A and A3B proteins fused to Cas9 for finding inhibitory effects of Ade1 and Ade2. Chimera proteins were made by swapping the loop-1 and loop-7 of the catalytic domain of A3A, A3B and A3G. Crystal structures of the ssDNA-A3A, ssDNA-A3Bloop-1chimera, and ssDNA-A3Gctd complexes are available to see how residues located in these loops recognize TC or CCC target sequences, and the effect of the loop swapping (Figure S1) suggest that A3B loop1 is important for the inhibition by EBV-BORF2 (Ade1). It is plausible to assume that Ade1 may inhibit ssDNA binding of A3B by physically blocking the access to loop1. These results are helpful to design biochemical and structural studies to elucidate the mechanisms of A3B binding/inhibition by EBV-BORF2.

Main concern is that both ribonuclease reductase (Ade1 and Ade2) and Vif (Ade3 and Ade4) affect deamination efficiency by reducing the number of APOBEC3 molecules in nucleus or cells, respectively. If controlling the number of A3-CBE molecules is the mechanism to reduce off-target deaminations, other experimental conditions which affect the protein levels in cells should be tested with and without Ades.

Although the use of viral anti-APOBEC proteins is a new idea and this manuscript provide interesting observations, the authors do not provide thorough data to prove that the Ades are the good tool to control the off-target activity of A3-CBEs.

Points of concerns:

HIV-Vif and SIVmac239-Vif recruit cellular ubiquitin E3-ligase and lead APOBEC3G for proteasomal degradation. This mechanism has been extensively studied. Since Vif simply reduces the number of A3G or A3G-fused Cas9 molecules in cells, Vif is not a good candidate for reducing the off-target deamination specifically while keeping efficacy of on-target deamination.

The Inhibitory mechanism of Ades section is not really informative as the molecular mechanism of the inhibition by Ade1 and Ade2 are not revealed. Literatures have suggested that the relocalization of A3A and A3B from nucleus to cytosol is the mechanism by which viral ribonucleotide reductases (Ade1 and Ade2) act as counter-restriction against A3A and A3B. Whether Ade1 and Ade2 directly affect the catalytic activity of A3A or A3B should be tested by using in vitro catalytic assays.

Figure 2C show that the amounts of A3Bctd-CBE were less in the existence of Ade1 or Ade2 compared with no inhibitor. Experimental procedure and supplements do not describe the detail of transfections of plasmids or western blots. Is this whole cell lysate? Did the authors check relocalization of A3-CBEs by Ade1 or Ade2? Also, data are missing to show protein amounts of other CBE-Cas9s used in this study. Since the protein amount of Cas9-CBE in nucleus or cells is supposed to be a key to affect the deamination frequency by Ades in this study, protein amount should be monitored always.

Figure 3B shows that the Cas9-dependent off-target editing by A3A-CBE was inhibited by Ade1 that supposed not have significant inhibitory effect on A3A. This observation generates a question whether there are unknown mechanisms causes the inhibition by Ade1.

Dear editor and reviewers:

Thank you very much for your comments concerning our manuscript entitled “Inhibition of base editors with anti-deaminases derived from viruses (NCOMMS-21-26365)”. Those comments are all valuable and very helpful for revising and improving our paper, as well as the important guiding significance to our researches. We have revised the manuscript accordingly (with blue fonts in the text) and a detailed response to the reviewers’ comments has been provided below.

Responses to Reviewer’s Comments:

To Reviewer #1:

1. * The authors stated that the Ades can suppress both Cas9-dependent and -independent off-target DNA editing. However, the authors only selected one site (EMX1-1) with only two Cas9-dependent and two Cas9-independent off-target sites to evaluate in this manuscript, which is insufficient to support their strong conclusions (Fig. 3). The authors should evaluate additional sites to thoroughly evaluate this effect. Alternatively, the authors can re-focus the manuscript to not heavily rely on the ability of these inhibitors to change on- to off-target editing activities.

Response:

Thank you for your kind suggestion. More sites with Cas9-dependent (three OT sites of HEK site4) and Cas9-independent (four SaCas9 target sites) off-target editing have been evaluated, and the results have been added in Fig. 5C and S8 of the revised manuscript. In addition, we re-focus the manuscript to the exploration and application of Ades in the revised manuscript accordingly.

2. * There is a lack of no-treatment controls presented in all figures. I wonder whether certain base mutagenesis events in some figures are a result of “background mutations” in the cell; therefore, I strongly recommended the authors include a control group for each figure.

Response:

Thank you for your good suggestion. The no-treatment controls have been added in all figures of the revised manuscript accordingly.

3. * It seems that the Ades inhibited the activity of base editors, but did not completely negate all base editing activities (the base editors can still induce both on- and off-target edits at a low level even when treated with Ades), which may be a bottleneck for various application of Ades, especially CBE-based therapeutic applications. Thus, I recommend the authors discuss whether the editing efficiencies following treatment with Ades is safe or precise enough in the revised manuscript. Furthermore, I recommend the authors explore the combination of using both AcrIIA5 and Ades and evaluate whether this would enhance the suppression of base editors.

Response:

Thank you for your good suggestion. The results have been added in Fig. S11 and discussed in

310-316 of the revised manuscript accordingly, which shown that combination of using both AcrIIA5 and Ades further enhance the suppression of base editors

4. * The authors declared that “The inhibitory ranges of Ade were further expanded to CGBE, A&CBE, A&CGBE, rA1-CBE and even ABE8e by rationally engineering the deaminases domain.” in the Abstract. However, I believe that this statement is an over-exaggeration as the Ades only modestly decrease the activities of other base editors according to the data shown in the manuscript. Secondly, certain deaminase activities are not affected by Ades at particular target sites (such as ABE8e activity in Fig. 4H). Therefore, I recommend the authors soften their conclusions.

Response:

Thank you for your kind suggestion. This conclusion has been softened in line 40-42 of the revised manuscript accordingly.

5. * The results in Fig. 4B are very interesting and exciting, which showed that variants of rA1, A3A, and A3Bctd significantly altered the purity of CGBEs. However, the authors only evaluated two sites, which is insufficient to draw the conclusion that A3A-N57G, A3Bctd-R211A, and A3Bctd-R211K are the best candidates for generating C-to-G base editing (for example, C-to-G efficiency of A3Bctd-R211A is lower than A3Bctd and A3Bctd-R211K at ABE site7 in Fig. S5, and C-to-G efficiency of A3Bctd-R211K is lower than A3Bctd-R211A and A3Bctd-VHM). Thus, I recommend the authors evaluate additional sites and also test the editing effects of using A3A-CBE (“wild-type” version) in Fig. 4B. Moreover, I am confused as to why most of the variants engineered for other purposes (such as A3A-N57G [PMID: 30059493] is engineered for preferentially editing “TC” motifs, and A3Bctd-VHM [PMID:32721385] was engineered for reducing the Cas9-independent off-target effects) showed significantly increased C-to-G editing. I recommend the authors discuss their hypothesis on how these mutations affect C-to-G editing in the manuscript.

Response:

Thank you for your kind suggestion. The evaluation of additional sites (RNF2, ABE site7, ABE site8 and HEK site2) and test the editing effects of using A3A-CBE have been added in Fig. S7 and line 186-190 of the revised manuscript. These results suggest that the engineering deaminase is an efficient method to improve C-to-G editing efficiency, consistent with a previous report of rA1-R33A^{1,2,3}.

Although the mechanism of improved C-to-G efficiency by engineering deaminases is not clear, we hypothesis that narrowing the deamination window of deaminase might increase the C-to-G editing outcome, which has been discussed in line 191-193 of the revised manuscript accordingly.

6. * I recommend the authors highlight the species from which the anti-deaminases originated in

the manuscript. Furthermore, to measure the cross-species activity of the Ades, I recommend the authors test whether certain Ades could inhibit the activities of deaminases from other species.

Response:

Thank you for your kind suggestion. The anti-deaminases originated from viruses have been highlighted in line 96-97 accordingly.

In addition, the inhibit activities of three deaminases from other species, including RmA3Bctd (rhesus monkey), mA3CDA1 (mouse) and SsA3Bctd (*Sus scrofa*) were tested. They did not show inhibition effect on those deaminases except for Ade1 which inhibited RmA3Bctd-CBE slightly, suggesting the Ades were species specific and mainly evolved to inhibited human A3 deaminases. The results have been added in Fig. S1 and line 110-114 of the revised manuscript accordingly.

7. * It seems that the ratio of Ade to CBE at 3:1 is not a maximal concentration for inhibition (Fig. 1E and Fig. 1F). I wonder if the author have tested higher doses of Ade1 and Ade2 for inhibition, and if so, please note this in the manuscript or methods?

Response:

Thank you for your kind suggestion. The ratios of Ade to CBE ranging from 1:4 to 6:1 were tested, which have been added in Fig. 1E-1G of the revised manuscript accordingly.

8. * Line 32: “Cytosine base editors (CBEs), comprising cytidine deaminases fused to the N-terminus of Cas9 nickase (nCas9)” This sentence is inaccurate. Although the original version of CBEs, such as BE3, BE4 used cytidine deaminases fused to the N-terminus of Cas9 nickase, there are many newer variants with CBEs fused to other regions. I recommend the authors revise this sentence.

Response:

Thank you for your kind suggestion. The sentence has been revised in line 32 of the revised manuscript accordingly.

9. * Line 34: I recommend the authors revise the sentence “However, the overexpression of cytidine deaminases leads to both Cas9-dependent off-target activities and Cas9-independent off-target activities...” since only the “Cas9-independent off-target activities” are induced by “overexpression of cytidine deaminases”.

Response:

Thank you for your kind suggestion. The sentence has been revised in line 34-35 of the revised manuscript accordingly.

10. * Line 40: For the sentence “Moreover, both Cas9-dependent off-target activity and Cas9-independent off-target activity of CBEs dramatically decreased at the presence of Ades, while the latter cannot be inhibited by the anti-CRISPRs (Acrs)”, I believe the data does not reflect this phenomenon in the manuscript. According to Fig 3C, the AcrIIA5 (green column) can

significantly decrease Cas9-independent off-target editing. I recommend the authors perform a statistical significance test comparing the AcrIIA5 and Ade1/2 to demonstrate the superiority of Ades for reducing Cas9-independent off-target editing.

Response:

Thank you for your good suggestion. The sentence has been revised in line 42-44 of the revised manuscript accordingly. The statistical significance test comparing the AcrIIA5 and Ade1/2 has been added in Fig. 5C accordingly.

11. * Line 97: The authors mentioned “BE4max” so I recommend the authors cite the relevant manuscript (PMID: 29813047).

Response:

Thank you for your good suggestion. The relevant reference has been cited in line 98 of the revised manuscript accordingly.

12. * Line 102: Please check if “(Fig. 3C and 3D).” should be “(Fig. 1C and 1D).”

Response:

Thank you for pointing this out. It has been revised in line 103 of the revised manuscript accordingly.

13. * Line 127: “Ade1only” should be “Ade1 only”.

Response:

Thank you for pointing this out. It has been revised in line 133 of the revised manuscript accordingly.

14. * Line 148, “A3-chineras” should be “A3-chimeras”.

Response:

Thank you for pointing this out. It has been revised in line 154 of the revised manuscript accordingly.

15. * Line 162: The “Sa site” should be detailed clarified, such as “SaCas9-induced R-loop region” or others.

Response:

Thank you for your good suggestion. It has been revised in line 236-237 of the revised manuscript accordingly.

16. * Lines 181-184: The sentence “However, there is no natural inhibitor available for them, they cannot be inhibited by the Ades ... due to the low sequence homology with A3 deaminases”. The use of competing pronouns makes this sentence extremely difficult to comprehend. I recommend the authors revise and clear up this sentence.

Response:

Thank you for your good suggestion. It has been revised in line 171-172 of the revised

manuscript accordingly.

17. * Line 301: The fonts of all “#” should be identical.

Response:

Thank you for pointing this out. It has been revised in line 334 of the revised manuscript accordingly.

18. * Lines 326 and Line 327: “200ng” should be “200 ng”.

Response:

Thank you for pointing this out. It has been revised in line 372 of the revised manuscript accordingly.

19. * Lines 352-481: The format of all references should be identical. Some references lack a volume number and/or page number (such as Ref. 5, 13, 14, 16 etc.). Some references lack or use an incorrect journal name (such as Ref. 50, 55).

Response:

Thank you for your good suggestion. The format of references has been revised accordingly.

To Reviewer #2:

1. * HIV-Vif and SIVmac239-Vif recruit cellular ubiquitin E3-ligase and lead APOBEC3G for proteasomal degradation. This mechanism has been extensively studied. Since Vif simply reduces the number of A3G or A3G-fused Cas9 molecules in cells, Vif is not a good candidate for reducing the off-target deamination specifically while keeping efficacy of on-target deamination.

Response:

Thank you for your good suggestion. The Vif might not be a good candidate for specifically reducing the off-target deamination while keeping efficacy of on-target deamination. However, we demonstrated that both Ade3 and Ade4 can inhibit A3G-CBE activity, holding the potential to act as off switches to regulate CBE in specific applications. We have discussed this in line 289-293 of the revised manuscript accordingly.

2. * The Inhibitory mechanism of Ade3 section is not really informative as the molecular mechanism of the inhibition by Ade1 and Ade2 are not revealed. Literatures have suggested that the relocalization of A3A and A3B from nucleus to cytosol is the mechanism by which viral ribonucleotide reductases (Ade1 and Ade2) act as counter-restriction against A3A and A3B. Whether Ade1 and Ade2 directly affect the catalytic activity of A3A or A3B should be tested by using in vitro catalytic assays.

Response:

Thank you for your good suggestion. The results shown that both Ade1 and Ade2 caused relocalization of A3Bctd- and A3A-CBEs from nucleus to perinuclear bodies, which have been

added in Fig. 2H-2G, Fig S4 and line 157-164 of the revised manuscript accordingly.

In addition, it was reported that Ade1 (EBV-BORF2) directly inhibits A3B catalytic activity by *in vitro* deaminase activity assay⁴. Accordingly, we tried to perform *in vitro* catalytic assays of A3A, but failed to acquire their proteins. The A3A construct is highly mutagenic in *E. coli*.

Although we failed to perform *in vitro* catalytic assays, our *in vivo* results suggested that both Ade1 and Ade2 inhibit deaminase catalytic domain, and Ade1 specifically interacts with L1 and L7 region of A3Bctd. In order to better understand our results, we have discussed and speculated it in line 296-301 of the revised manuscript accordingly.

3. * Figure 2C show that the amounts of A3Bctd-CBE were less in the existence of Ade1 or Ade2 compared with no inhibitor. Experimental procedure and supplements do not describe the detail of transfections of plasmids or western blots. Is this whole cell lysate? Did the authors check relocalization of A3-CBEs by Ade1 or Ade2? Also, data are missing to show protein amounts of other CBE-Cas9s used in this study. Since the protein amount of Cas9-CBE in nucleus or cells is supposed to be a key to affect the deamination frequency by Ades in this study, protein amount should be monitored always.

Response:

Thank you for your good suggestion. The details of transfections of plasmids and western blots were added in line 349-354 of the revised manuscript accordingly. The whole cell lysate was used for immunoblot.

The results of relocation of A3Bctd-CBE and A3A-CBE have been added in Fig. 2H-2G, Fig S4 and line 157-164 of the revised manuscript.

The western blots results of other A3-CBEs with or without Ades have been added in Fig. S2 of the revised manuscript accordingly.

4. * Figure 3B shows that the Cas9-dependent off-target editing by A3A-CBE was inhibited by Ade1 that supposed not have significant inhibitory effect on A3A. This observation generates a question whether there are unknown mechanisms causes the inhibition by Ade1.

Response:

Thank you for your kind suggestion. This observation may be explained by the result that Ade1 can relocate A3A-CBE from nucleus to perinuclear bodies, which have been discussed in line 246-251 of the revised manuscript accordingly.

References

1. Kurt IC, *et al.* CRISPR C-to-G base editors for inducing targeted DNA transversions in human cells. *Nature biotechnology* **39**, 41-46 (2021).
2. Koblan LW, *et al.* Efficient C•G-to-G•C base editors developed using CRISPRi

screens, target-library analysis, and machine learning. *Nature biotechnology*, (2021).

3. Yuan T, *et al.* Optimization of C-to-G base editors with sequence context preference predictable by machine learning methods. *Nature communications* **12**, 4902 (2021).

4. Cheng AZ, *et al.* Epstein-Barr virus BORF2 inhibits cellular APOBEC3B to preserve viral genome integrity. *Nature microbiology* **4**, 78-88 (2019).

Reviewers' Comments:

Reviewer #1:

Remarks to the Author:

The authors have addressed all my concerns.

Reviewer #2:

Remarks to the Author:

The authors have answered this reviewer's concerns by adding new experiments. The immunofluorescence experiments showed that some A3A-CBE and A3Bctd-CBE were located in perinuclear region when they were co-expressed with Ade1 or Ade2, although significant amounts of CBEs were also found in nucleus. The revised manuscript focused on the inhibition of total deaminase activity of CBEs rather than specific inhibition of off-target deamination, which reflects the experimental results presented in the manuscript better than the previous version.

Comments:

84 In this study, we reported the discovery of CBE-inhibiting proteins derived from viruses.

In my opinion, "discovery" is not appropriate because CBE inhibiting proteins had been reported.

166 and relocation. Specifically, Ade1 interacts with the L1 and L7 regions, and both Ade1 and Ade2 relocate A3-CBEs from the nucleus to the perinuclear region.

Fig. 2H, 2G and S4 show similar localization patterns of A3A-CBE and A3Bctd-CBE with Ade1 and Ade2. In particular, A3A-CBE+Ade1 and A3A-CBE+Ade2 showed very similar localization patterns, yet Ade1 did not inhibit on-target deamination by A3A-CBE (Fig. 1). These results suggest that relocation of CBEs was not the major mechanism of inhibition by Ade1 and Ade2.

The authors response in the rebuttal letter "In addition, it was reported that Ade1 (EBV-BORF2) directly inhibits A3B catalytic activity by in vitro deaminase activity assay⁴. Accordingly, we tried to perform in vitro catalytic assays of A3A, but failed to acquire their proteins. The A3A construct is highly mutagenic in *E. coli*."

This reviewer thinks that in vitro deaminase assays of Ade2 against A3A and A3Bctd (or A3A-CBE and A3Bctd-CBE) would provide essential information for the proposed inhibition methods of base editors. In vitro deaminase assays for A3A have been reported by quite a few laboratories. The authors would be able to refer the methods described in Byeon et al., *Nat Commun.* 2013: PMID: 23695684 .

281 with various types of BEs, including CGBE, A&CBE, A&CGBE, rA1-CBE and even ABE8e. We found that Ades were safer off switches for CBEs than conventional Acrs because they completely shut down the activities of cytidine deaminase, the effector domain in CBEs.

Experimental results in this study indicated that Ades did not completely shut down the activities of CBEs.

Dear editor and reviewers:

Thank you very much for your comments concerning our manuscript entitled “Inhibition of base editors with anti-deaminases derived from viruses (NCOMMS-21-26365A)”. Those comments are all valuable and very helpful for revising and improving our paper, as well as the important guiding significance to our researches. We have revised the manuscript accordingly (with blue fonts in the text) and a detailed response to the reviewers’ comments has been provided below.

Responses to Reviewer’s Comments:

To Reviewer #1:

1. * The authors have addressed all my concerns.

Response:

Thank you for your kind suggestion.

To Reviewer #2:

1. * 84 In this study, we reported the discovery of CBE-inhibiting proteins derived from viruses. In my opinion, “discovery” is not appropriate because CBE inhibiting proteins had been reported.

Response:

Thank you for your good suggestion. It has been revised in line 84-85 of the revised manuscript accordingly.

2. * 166 and relocation. Specifically, Ade1 interacts with the L1 and L7 regions, and both Ade1 and Ade2 relocate A3-CBEs from the nucleus to the perinuclear region.

Fig. 2H, 2G and S4 show similar localization patterns of A3A-CBE and A3Bctd-CBE with Ade1 and Ade2. In particular, A3A-CBE+Ade1 and A3A-CBE+Ade2 showed very similar localization patterns, yet Ade1 did not inhibit on-target deamination by A3A-CBE (Fig. 1). These results suggest that relocation of CBEs was not the major mechanism of inhibition by Ade1 and Ade2.

Response:

Thank you for your good suggestion. Both Ade1 and Ade2 relocated A3-CBEs from the nucleus to the perinuclear region, suggesting that the relocation of A3-CBEs was not the major mechanism of different inhibition by Ade1 and Ade2, which has been discussed in line 164-166 of the revised manuscript. To clarify the mechanism of inhibition, the *in vitro* deaminase activity assay was carried out, and the results has been added in Fig S5 and discussed in line 167-173 of the revised manuscript accordingly.

3. * The authors response in the rebuttal letter “In addition, it was reported that Ade1(EBV-BORF2) directly inhibits A3B catalytic activity by in vitro deaminase activity assay⁴. Accordingly, we tried

to perform in vitro catalytic assays of A3A, but failed to acquire their proteins. The A3A construct is highly mutagenic in *E. coli*.”

This reviewer thinks that in vitro deaminase assays of Ade2 against A3A and A3Bctd (or A3A-CBE and A3Bctd-CBE) would provide essential information for the proposed inhibition methods of base editors. In vitro deaminase assays for A3A have been reported by quite a few laboratories. The authors would be able to refer the methods described in Byeon et al., *Nat Commun.* 2013: PMID: 23695684 .

Response:

Thank you for your kind suggestion. The *in vitro* deaminase activity assay was carried out. As a result, Ade1 potently inhibited A3Bctd and Ade2 moderately inhibit both A3A and A3Bctd, which was consistent with the observations of A3-CBEs in human cells. Interestingly, the Ade1 also showed slight inhibition to A3A *in vitro*, while no obvious inhibition of A3A-CBE was observed in human cells. The results have been added in Fig S5 and discussed in line 167-173 of the revised manuscript accordingly.

4. * 281 with various types of BEs, including CGBE, A&CBE, A&CGBE, rA1-CBE and even ABE8e. We found that Ades were safer off switches for CBEs than conventional Acrs because they completely shut down the activities of cytidine deaminase, the effector domain in CBEs. Experimental results in this study indicated that Ades did not completely shut down the activities of CBEs.

Response:

Thank you for your kind suggestion. The “completely shut down the activities” has been substituted by “directly inhibit the activities” in the line 288-289 of the revised manuscript accordingly.

Reviewers' Comments:

Reviewer #2:

Remarks to the Author:

The authors have addressed all of my concerns.

Responses to Reviewer's Comments:

To Reviewer #2:

1. * The authors have addressed all of my concerns.

Response:

Thank you for your kind suggestion.